# *daf-16/*FOXO blocks adult cell fate in *Caenorhabditis elegans* dauer larvae via *lin-41/* TRIM71

**Matthew J. Wirick**[1]☉, **Allison R. Cale**[2,3]☉, **Isaac T. Smith**[2], **Amelia F. Alessi**[4], **Margaret R. Starostik**[4], **Liberta Cuko**[2], **Kyal Lalk**[2], **Mikayla N. Schmidt**[2], **Benjamin S. Olson**[2], **Payton M. Salomon**[2], **Alexis Santos**[2], **Axel Schmitter-Sánchez**[2], **Himani Galagali**[4], **Kevin J. Ranke**[2], **Payton A. Wolbert**[2,3], **Macy L. Knoblock**[2,3], **John K. Kim**[4], **Xantha Karp**[1,2]*

1 Biochemistry, Cell & Molecular Biology Program, Central Michigan University, Mt Pleasant, Michigan, United States of America, 2 Department of Biology, Central Michigan University, Mt Pleasant, Michigan, United States of America, 3 Department of Chemistry and Biochemistry, Central Michigan University, Mt Pleasant, Michigan, United States of America, 4 Department of Biology, Johns Hopkins University, Baltimore, Maryland, United States of America

☉ These authors contributed equally to this work.
* karp1x@cmich.edu

**Data Availability Statement:** All relevant data are within the manuscript and its Supporting Information files except that mRNA-seq data have been deposited in NCBI under GEO accession

## Abstract

Many tissue-specific stem cells maintain the ability to produce multiple cell types during long periods of non-division, or quiescence. FOXO transcription factors promote quiescence and stem cell maintenance, but the mechanisms by which FOXO proteins promote multipotency during quiescence are still emerging. The single FOXO ortholog in *C. elegans*, *daf-16*, promotes entry into a quiescent and stress-resistant larval stage called dauer in response to adverse environmental cues. During dauer, stem and progenitor cells maintain or re-establish multipotency to allow normal development to resume after dauer. We find that during dauer, *daf-16/*FOXO prevents epidermal stem cells (seam cells) from prematurely adopting differentiated, adult characteristics. In particular, dauer larvae that lack *daf-16* misexpress collagens that are normally adult-enriched. Using *col-19p*::*gfp* as an adult cell fate marker, we find that all major *daf-16* isoforms contribute to opposing *col-19p*::*gfp* expression during dauer. By contrast, *daf-16(0)* larvae that undergo non-dauer development do not misexpress *col-19p*::*gfp*. Adult cell fate and the timing of *col-19p*::*gfp* expression are regulated by the heterochronic gene network, including *lin-41* and *lin-29*. *lin-41* encodes an RNA-binding protein orthologous to LIN41/TRIM71 in mammals, and *lin-29* encodes a conserved zinc finger transcription factor. In non-dauer development, *lin-41* opposes adult cell fate by inhibiting the translation of *lin-29*, which directly activates *col-19* transcription and promotes adult cell fate. We find that during dauer, *lin-41* blocks *col-19p*::*gfp* expression, but surprisingly, *lin-29* is not required in this context. Additionally, *daf-16* promotes the expression of *lin-41* in dauer larvae. The *col-19p*::*gfp* misexpression phenotype observed in dauer larvae with reduced *daf-16* requires the downregulation of *lin-41*, but does not require *lin-29*. Taken together, this work demonstrates a novel role for *daf-16/*FOXO as a heterochronic gene that

number GSE179166. Processed data and scripts used for analysis are available at https://github.com/starostikm/DAF-16.

**Funding:** This work was supported by R01GM118875 to JKK from the National Institutes of Health, https://www.nih.gov; R01GM129301 to JKK from the National Institutes of Health, https://www.nih.gov; R15GM117568 to XK from the National Institutes of Health, https://www.nih.gov; CAREER 1652283 to XK from the National Science Foundation https://www.nsf.gov. The funders had nobrole in study design, data collection and analysis, decision to publish, or preparation of the manuscript.

**Competing interests:** The authors have declared that no competing interests exist.

promotes expression of *lin-41/*TRIM71 to contribute to multipotent cell fate in a quiescent stem cell model.

## Author summary

In adults and juveniles, tissue-specific stem cells divide as needed to replace cells that are lost due to injury or normal wear and tear. Many stem cells spend long periods of time in cellular quiescence, or non-division. During quiescence, stem cells remain multipotent, where they retain the ability to produce all cell types within their tissue. In this study, we define a new role for the FOXO protein DAF-16 in promoting multipotency during the quiescent *C. elegans* dauer larva stage. *C. elegans* larvae enter dauer midway through development in response to adverse environmental conditions. Epidermal stem cells are multipotent in *C. elegans* larvae but differentiate at adulthood, a process controlled by the "heterochronic" genes. We found that *daf-16* blocks the expression of adult cell fate specifically in dauer larvae by promoting the expression of the heterochronic gene *lin-41*. *lin-41* normally blocks adult fate by repressing the expression of another heterochronic gene, *lin-29*, but surprisingly, *lin-29* is not needed for the expression of adult cell fate in this context. These findings may be relevant to mammals where the orthologs of *daf-16* and *lin-41* are important in stem cell maintenance and opposing differentiation.

## Introduction

Tissue-specific stem cells divide as needed to replenish cells lost due to injury or normal wear and tear. Many stem cell types retain the capacity to produce multiple cell types during lengthy periods of quiescence, or non-division, and increased cell division can lead to compromised multipotency and stem cell maintenance [1]. However, the connections between quiescence and multipotency are incompletely understood. *Caenorhabditis elegans (C. elegans)* development can be interrupted by a quiescent and stress-resistant stage called dauer in response to adverse environmental conditions [2]. Stem cell-like progenitor cells maintain or re-establish multipotency during dauer, a situation analogous to mammalian stem cells [3–5].

Dauer formation is regulated by three major signaling pathways [6,7]. One of these pathways is insulin/IGF signaling, where favorable environmental cues lead to the production of multiple insulin-like peptides in sensory neurons. [8–13]. These signals are released and then received in multiple tissues where insulin signaling blocks the activity of the downstream DAF-16/FOXO transcription factor [14–16]. In adverse environments, DAF-16 is active and regulates the expression of genes that promote dauer formation [6,10].

The role of the DAF-16/FOXO transcription factor in promoting dauer formation is analogous to the role of the FOXO proteins in promoting quiescence in mammalian stem cells [17]. In addition, DAF-16/FOXO is required for stem cell maintenance and the ability of stem/progenitor cells to differentiate into the correct cell types in both systems [5,10,18–22]. For example, in *C. elegans* dauer larvae, *daf-16* is required in a set of multipotent vulval precursor cells to block EGFR/Ras and LIN-12/Notch signaling from prematurely specifying vulval cell fates [5].

Another group of progenitor cells that must remain quiescent and multipotent during dauer is the lateral hypodermal seam cells that make up part of the worm skin. Seam cells undergo self-renewing divisions at each larval stage and then terminally differentiate at

adulthood [23]. Larval vs. adult seam cell fate is regulated by a network of heterochronic genes [24]. In general, heterochronic transcription factors and RNA-binding proteins that specify early cell fates are expressed early in development. These early cell fate-promoting factors are then downregulated by microRNAs in order to allow progression to later cell fates [25]. This pathway culminates with the expression of the LIN-29 transcription factor. LIN-29 is the most downstream regulator of adult cell fate and directly activates the expression of adult-specific collagens such as *col-19* [26–29]. LIN-29 protein is not expressed in the hypodermis until late in larval development due to the combined action of the early-promoting heterochronic genes *lin-41* and *hbl-1* [27,30,31]. *lin-41* encodes an RNA-binding protein that binds to the *lin-29* mRNA and blocks translation of the *lin-29a* isoform [30,32]. *lin-29* is also required for adult cell fate in post-dauer animals. However, many genes that act earlier in the heterochronic pathway are dispensable for post-dauer development, suggesting that the regulation of adult cell fate differs in continuous and dauer life histories [3].

Here, we examine the role of *daf-16*/FOXO in maintaining multipotent seam cell fate during dauer. We find that *daf-16* is required to block the expression of multiple adult-enriched collagens during dauer, including *col-19*, defining a novel role for *daf-16* as a dauer-specific heterochronic gene. Using the adult cell fate marker *col-19p*::*gfp* as a readout, we find that *daf-16* acts via *lin-41* to regulate adult cell fate. Surprisingly, *lin-29* plays at most a minor role in the regulation of *col-19p*::*gfp* during dauer. mRNA-seq experiments identified 3603 genes that are regulated by *daf-16* during dauer, including 112 transcription factors that may mediate repression of adult-enriched collagens during dauer downstream of *daf-16*.

## Results

### *daf-16* blocks adult-specific collagen expression in dauer larvae

Progenitor cells remain multipotent throughout dauer diapause, and *daf-16* is required to maintain or re-establish multipotency in the vulval precursor cells (VPCs) of dauer larvae [5]. We asked whether *daf-16*/FOXO might also promote multipotency in lateral hypodermal seam cells, another multipotent progenitor cell type. The well-characterized adult cell-fate marker *col-19p*::*gfp*, where GFP expression is driven by the adult-specific *col-19* promoter, is expressed in differentiated seam cells and the hyp7 syncytium in adult worms but is not expressed in multipotent seam cells or hyp7 in larvae [28,33]. To determine if *daf-16*/FOXO promotes multipotent, larval seam cell fate during dauer, we assayed larvae with reduced or absent *daf-16*, termed *"daf-16(-)"*, for precocious expression of *col-19p*::*gfp*. Specifically, we tested two *daf-16* null alleles, *daf-16(mu86)* and *daf-16(mgDf50)*, as well as *daf-16(RNAi)* dauer larvae. As *daf-16(-)* larvae are normally unable to enter dauer even in dauer-inducing environmental conditions (i.e., dauer defective), we took advantage of the temperature-sensitive dauer-constitutive allele *daf-7(e1372)* to enable dauer formation in *daf-16(-)* larvae. When grown at restrictive temperatures, both *daf-16(-); daf-7(-)* and control *daf-7(-)* larvae enter dauer even in favorable environments [34,35]. For simplicity, *daf-7(e1372)* is not always mentioned, but this allele was present in all *daf-16(-)* and control dauer larvae (see S1 Table for a complete list of strains used in this study). We found that *daf-16(-)* dauer larvae displayed penetrant *col-19p*::*gfp* expression in seam cells and hyp7 (Fig 1A). *col-19p*::*gfp* was expressed throughout the hypodermis, but was consistently brighter in the posterior third of the worm, gradually dimming toward the anterior. These findings indicate that *daf-16* is necessary to block *col-19p*::*gfp* expression during dauer. For the remainder of the paper, *"daf-16(0)"* is used to indicate the *daf-16(mgDf50)* null allele, a large deletion that removes most of the *daf-16* coding sequence, including the DNA-binding domain [15].

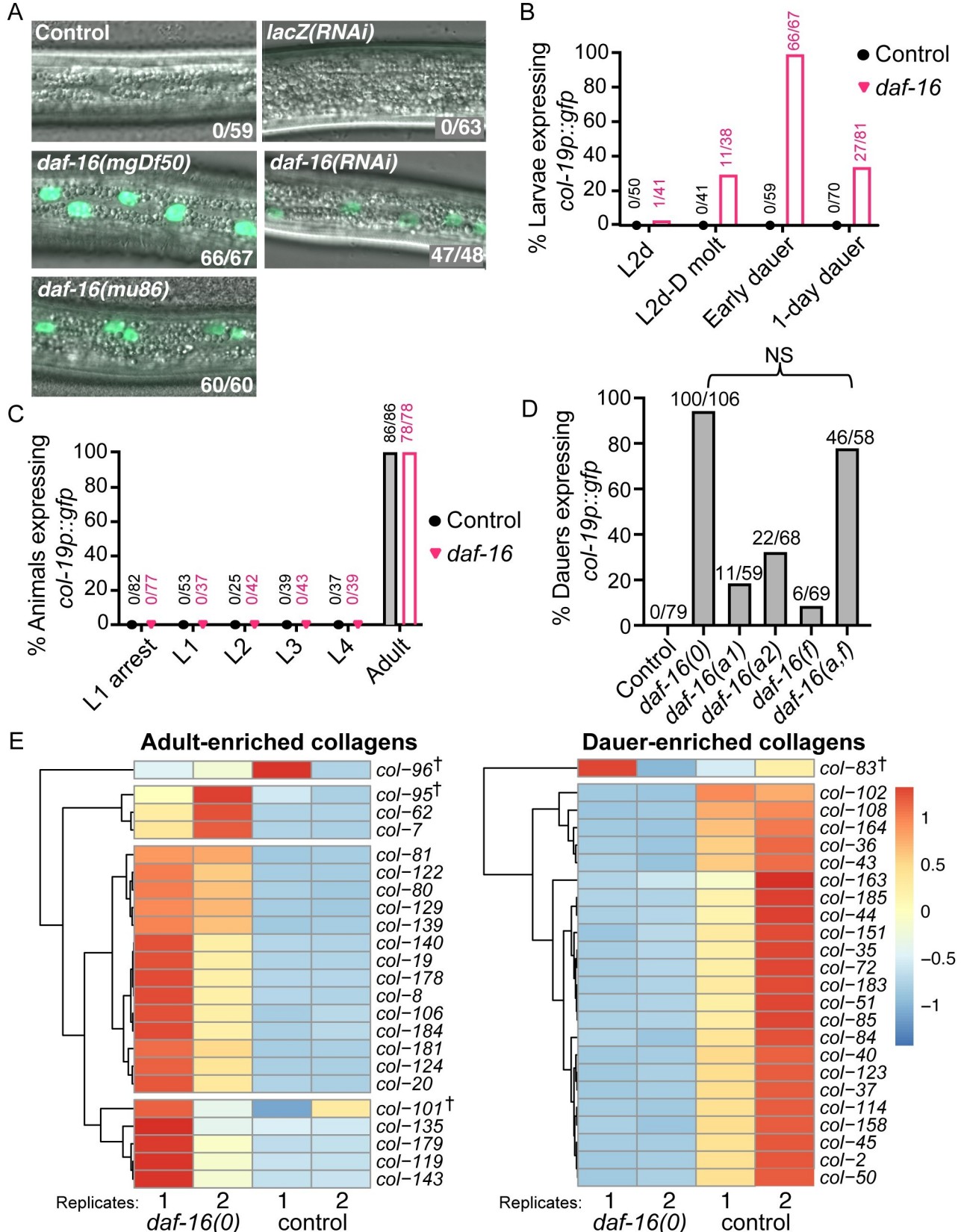

**Fig 1. *daf-16* is required to prevent precocious adoption of adult cell fate during dauer.** (A) Merged DIC and fluorescence images taken with a 63x objective displaying a portion of the lateral hypodermis in dauer larvae. Unless otherwise specified, the background for all strains in this and subsequent figures was *daf-7(e1372); col-19p::gfp*. (See S1 Table for full genotypes). Compromised *daf-16* activity due to either of two null alleles (left) or RNAi (right) resulted in *col-19p::gfp* expression. Numbers indicate larvae expressing *col-19p::gfp* over the total number of dauer larvae scored. (B) Percent of *daf-7* (control) and *daf-16(0); daf-7* larvae that expressed *col-19p::gfp* before dauer formation and over time in dauer. GFP expression began during the molt, peaked soon after entry into dauer, and declined thereafter. (C) Percent of wild-type (control) and *daf-16(0)* animals that expressed *col-19p::gfp* at different stages. These worms were wild-type for *daf-7* and therefore did not enter L2d or dauer. Larvae at L1 arrest were hatched in the absence of food and were monitored for GFP expression over seven days. (D) Percent of dauer larvae lacking some or all *daf-16* isoforms that expressed any hypodermal *col-19p::gfp* (see S1 Table for full genotypes). Note that *daf-16(a,f)* displayed decreased expressivity compared to *daf-16(0)*; the dauer larvae expressing *col-19p::gfp* were noticeably dimmer than in the null allele (S2 Fig). NS: p = 0.55, Fisher Exact Test. (E) Heat maps depicting relative expression of adult-enriched collagens (left) or dauer-enriched collagens (right) in *daf-16(0); daf-7* dauer larvae or *daf-7* (control) dauer larvae from mRNA-seq data. The row-normalized reads for each of two biological replicates per strain are depicted. These collagens were expressed at significantly different levels (FDR ≤ 0.05) in *daf-16(0)* vs control dauer larvae in all cases except those collagens indicated by †. Adult and dauer-enriched collagens were identified using modENCODE data (S3 Table). Adult-enriched collagens were upregulated in *daf-16(0)* dauer larvae whereas dauer-enriched collagens were downregulated.

Wild type *daf-16*/FOXO promotes dauer formation largely in parallel to the DAF-3/SMAD-DAF-5/Ski complex [6,7]. To test whether this complex is also required to block *col-19p::gfp* expression during dauer, we examined *daf-5(0)* dauer larvae. Similar to *daf-16(-)* larvae, *daf-5(0)* larvae do not normally develop through dauer. To obtain *daf-5(0)* dauers, we used the *daf-2(e1370)* allele to drive dauer formation in the *daf-5(0)* background. [34,35]. We found that *daf-5(0); daf-2(e1370)* dauer larvae never expressed *col-19p::gfp* (S1 Fig). Therefore, while *daf-16* and *daf-5* act in parallel to promote dauer formation, the regulation of adult cell fate is specific to *daf-16*.

We next asked whether *daf-16*/FOXO regulates *col-19p::gfp* expression during stages other than dauer. First, we examined the timing of *col-19p::gfp* expression in larvae before, during, and after their entry into dauer. In wild-type animals, *col-19p::gfp* expression begins during the L4 molt, peaks soon after molting to adulthood, and then declines over time. Similarly, in *daf-16(0)* mutants, *col-19p::gfp* expression begins during the L2d-to-dauer molt, peaks soon after larvae enter dauer, and then declines over time in dauer (Fig 1B). Next, we asked whether larvae that develop continuously express precocious *col-19p::gfp*. For this experiment, we used larvae that were wild-type for *daf-7*. *col-19p::gfp* expression was never observed in *daf-16(0)* larvae during continuous development, and *col-19p::gfp* was expressed normally in adults (Fig 1C). Finally, we asked whether *daf-16(0)* larvae in L1 arrest express *col-19p::gfp*. Dauer shares some similarities with L1 arrest, which is a developmentally arrested and quiescent stage occurring when L1 larvae hatch in the absence of food [36]. *daf-16*/FOXO is important for L1 arrest; arrested *daf-16(0)* mutant larvae precociously initiate post-embryonic development including V lineage seam cell divisions [21]. However, we saw no precocious *col-19p::gfp* expression in these larvae (Fig 1C).

We next wondered which *daf-16* isoforms block *col-19p::gfp* expression during dauer. There are three major isoforms of *daf-16*: *a*, *b*, and *f* [15,16,37]. The *a* and *f* isoforms together are the major players with respect to the role of *daf-16* in longevity, dauer formation, and stress resistance [38]. Isoform *a* plays a larger role than *f*, but the *f* isoform still contributes because mutants lacking isoforms *a* and *f* display a stronger phenotype than mutants lacking either single isoform. By contrast, the *b* isoform is dispensable because animals lacking *a* and *f* isoforms were indistinguishable from null mutants for the phenotypes tested [38]. In contrast, the *b* isoform regulates neuronal morphology and function [39,40]. To determine which *daf-16* isoforms contribute to maintaining seam cell multipotency during dauer, we used the same isoform-specific alleles used by Chen et al. (2015) [38] to assess *col-19p::gfp* expression during dauer. Loss of only *daf-16a* produced a stronger phenotype than loss of only *daf-16f*, and loss of both *daf-16a* and *f* produced an even stronger phenotype than loss of either individual isoform examined, indicating that both isoforms contribute (Figs 1D and S2). Loss of *a* and *f*

together caused expression of *col-19p*::*gfp* at high penetrance (Fig 1D), however this expression was dimmer and in fewer cells than the expression observed with in the null background (S2 Fig). Therefore, since loss *of daf-16a/f* did not produce a phenotype as strong as that of the complete null, the *b* isoform must also play a role. Taken together, we conclude that all three isoforms of *daf-16* contribute to blocking *col-19p*::*gfp* expression during dauer.

To extend these findings beyond *col-19p*::*gfp* reporter expression, we performed mRNA-seq and differential gene expression analysis on *daf-16(0)* vs. control dauer larvae. Consistent with the data from *col-19p*::*gfp* experiments, we found that endogenous *col-19* was highly and significantly upregulated in *daf-16* mutants. Furthermore, 19/22 other adult-enriched collagens were also upregulated (Fig 1E and S3 Table). Similar to *col-19p*::*gfp*, most of these adult-enriched collagens appear to be expressed in seam cells and in hyp7 (S4 Table). Consistent with the upregulation of adult-enriched collagens, 23/24 dauer-enriched collagens were downregulated in *daf-16(0)* dauer larvae (Fig 1E), indicating an overall shift to genes required for adult cuticle formation in *daf-16(0)* dauer larvae.

In wild-type animals, the onset of *col-19p*::*gfp* expression coincides with several other characteristics of adult seam cell fate, including the production of adult alae, cell-cycle exit, and seam-cell fusion [23,24,41]. Although *daf-16(0)* dauer larvae express adult collagens, they do not display these other adult characteristics. Consistent with previous reports, we observed that *daf-16(0)* dauer larvae display dauer alae, which are distinct from adult alae [15,34]. Using the *wIs78* transgenic strain that expresses GFP in both seam cell nuclei *(scm*::*gfp)* and apical junctions *(ajm-1*::*gfp)* [42], we found that seam cells in *daf-16(0)* dauer larvae remained unfused (S3 Fig). Additionally, seam cells in *daf-16(0)* dauer larvae failed to maintain quiescence and underwent cell divisions (S3 Fig). These cell divisions were similar to the asymmetric divisions that normally occur during the L1, L3, and L4 stages (S3 Fig). Seam cell division does not occur during either adult or dauer stages in wild-type animals, however *daf-16*/FOXO is required for cell-cycle arrest in other contexts [21,22]. Therefore, during dauer, *daf-16*/FOXO both promotes seam cell quiescence and blocks adult cell fate, where its role in regulating seam cell fate is predominantly to control stage-specific collagen expression.

## *daf-16* promotes expression of *lin-41* during dauer

During continuous development, the timing of *col-19p*::*gfp* expression is regulated by the heterochronic gene network [24,28,33,43]. To determine how *daf-16* interacts with heterochronic genes, we began by asking whether heterochronic genes that oppose adult cell fate during continuous development might also be required to block *col-19p*::*gfp* expression during dauer. We focused on the three most downstream of these heterochronic genes, *lin-14*, *hbl-1*, and *lin-41* (Fig 2A). We found that knockdown of *lin-14* and *hbl-1* resulted in little to no *col-19p*::*gfp* expression in hypodermal cells whereas *lin-41* RNAi resulted in penetrant *col-19p*::*gfp* expression in lateral hypodermal cells during dauer (Fig 2B). Similar to *daf-16(RNAi)*, *col-19p*::*gfp* expression in *lin-41(RNAi)* dauer larvae was brightest in the posterior of the worm, and dim or absent in the anterior. However, overall *col-19p*::*gfp* expression was dimmer in *lin-41(RNAi)* dauer larvae compared to *daf-16(RNAi)* (S4 Fig). In addition, some *col-19p*::*gfp* expression was observed in non-hypodermal cell types in *hbl-1(RNAi)* and *lin-41(RNAi)* dauer larvae (S5 Fig).

Since both *daf-16* and *lin-41* block *col-19p*::*gfp* expression in dauer larvae, we asked whether *daf-16* regulates *lin-41* expression. We examined our mRNA-seq data and also performed qPCR on *daf-16(0)* and control dauer larvae. In these experiments, *lin-41* was downregulated approximately 2-fold in *daf-16(0)* dauers compared to control dauer larvae (Fig 3A). In contrast, there was no significant change in mRNA levels of the other core heterochronic genes (S6 Fig). Together, these data suggest that during dauer, *daf-16* specifically promotes *lin-41*

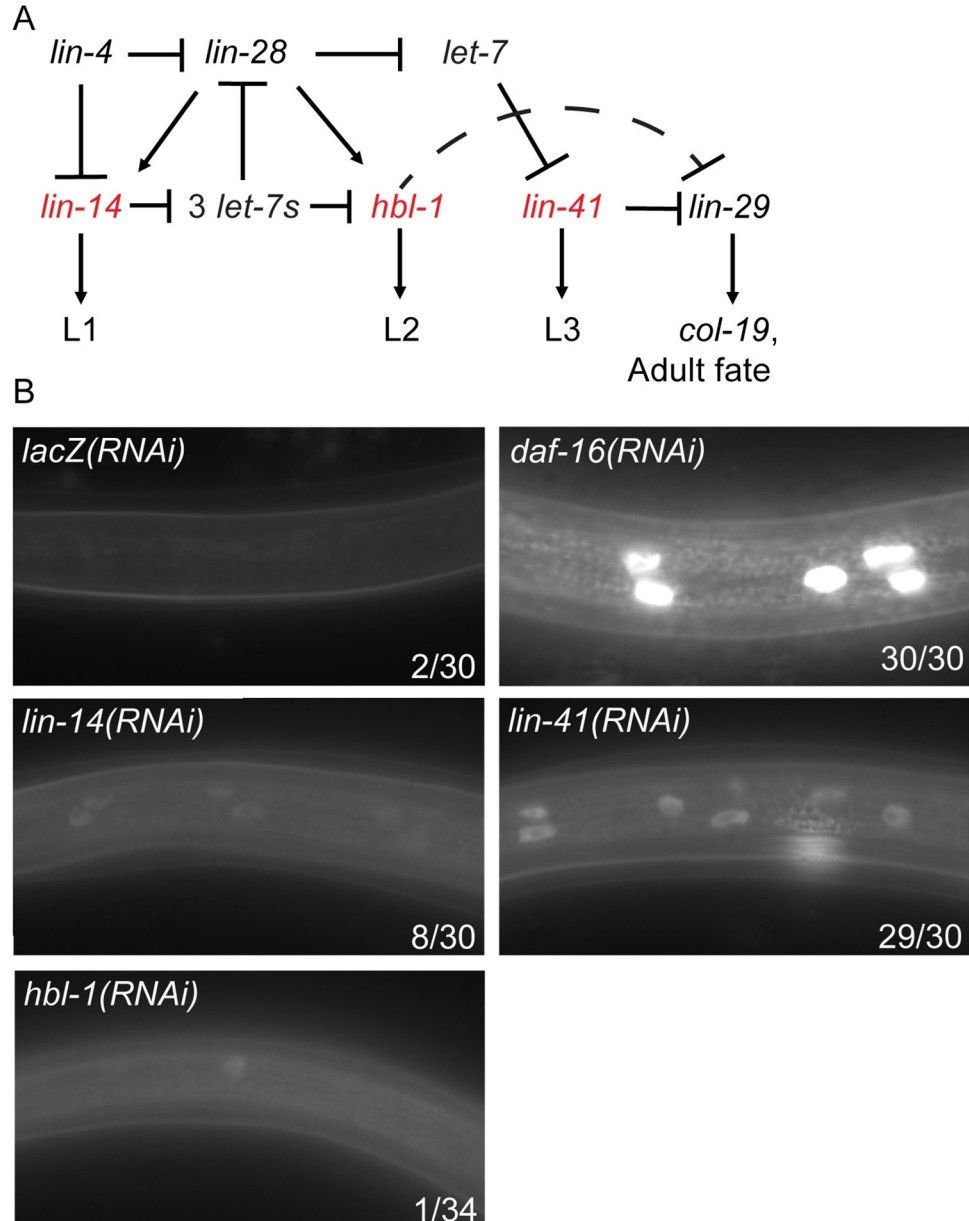

**Fig 2. *lin-41* is required to prevent precocious adoption of adult cell fate during dauer.** (A) A simplified diagram of the network of heterochronic genes that regulates stage-specific seam cell fate during continuous development. "3 *let-7s*" indicates the *let-7* family members, *mir-48*, *mir-84*, and *mir-241*. Genes in red are the most downstream positive regulators of larval cell fates [25]. (B) Precocious hypodermal expression of *col-19p::gfp* in dauer larvae in response to RNAi of heterochronic genes compared to the *lacZ* and *daf-16* RNAi controls. Images were taken with a 63x objective. Numbers indicate the number of larvae expressing any hypodermal *col-19p::gfp* over the total number of dauer larvae scored. *lin-14* RNAi induced only dim expression of *col-19p::gfp*, at low penetrance. *hbl-1* RNAi did not induce hypodermal expression of *col-19p::gfp*. *lin-41* RNAi caused highly penetrant expression of *col-19p::gfp* during dauer.

expression to block *col-19p::gfp* and adult cell fate, and that in *daf-16(-)* dauers, lower levels of *lin-41* lead to precocious *col-19p::gfp* expression.

To test whether reduced *lin-41* levels are required for precocious *col-19p::gfp* expression in *daf-16(0)* dauers, we performed *daf-16* RNAi on a strain in which *lin-41* is misexpressed. To do this, we took advantage of the *lin-41(xe8[Δ3'UTR])* allele which prevents the normal

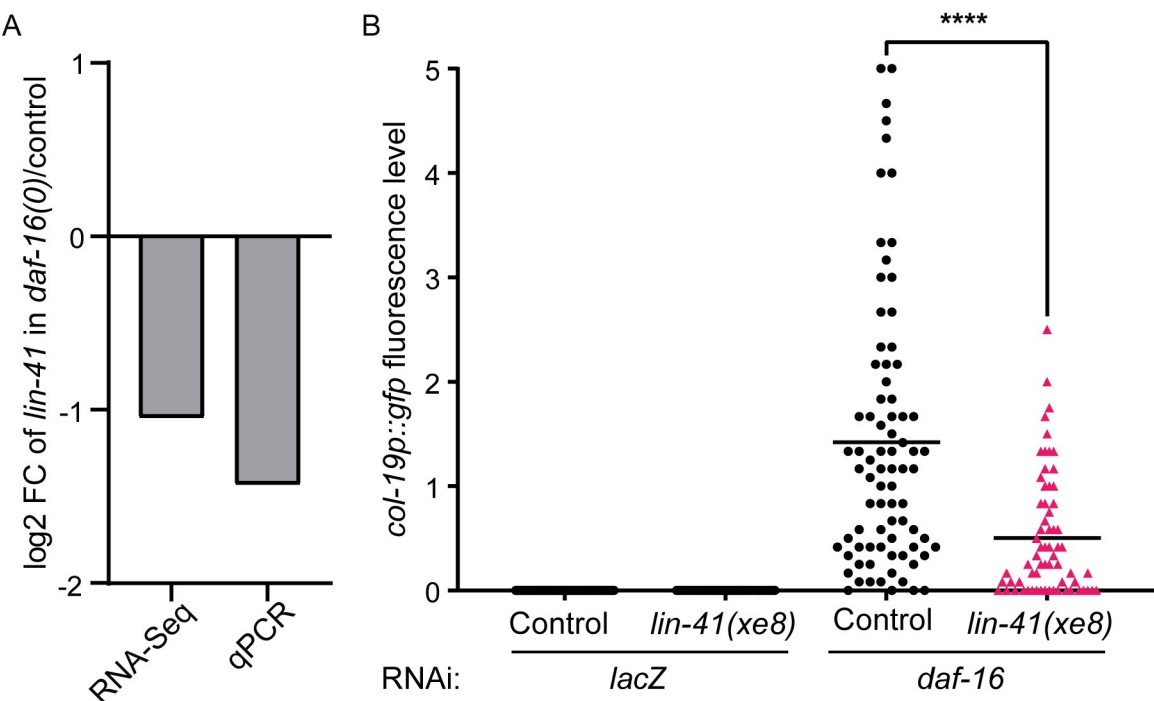

**Fig 3. *daf-16* promotes *lin-41* expression during dauer to regulate *col-19p::gfp*.** (A) *lin-41* was downregulated in *daf-16(0); daf-7* dauer larvae compared to *daf-7* (control) dauers, as measured by mRNA-seq (FDR < 0.05) and qPCR (p = 0.2, two-tailed Mann-Whitney test). At least two biological replicates per strain were used (see Methods). Raw qPCR data are shown in S5 Table. (B) Misexpression of *lin-41* caused by *lin-41(xe8[Δ3'UTR])* suppressed the *col-19p::gfp* phenotype in *daf-16(RNAi)* dauer larvae. Fluorescence levels were determined as described in S7 Fig. Each point represents an individual worm; horizontal lines indicate the mean fluorescence level. ****p<0.0001 (Two-tailed Mann-Whitney Test). n = 39–80; see S6 Table for complete underlying numerical data.

downregulation of *lin-41* in late larval stages and causes a strong gain-of-function phenotype [44,45]. We found the *col-19p::gfp* phenotype produced by *daf-16(RNAi)* was markedly suppressed in *lin-41(xe8)* dauer larvae, consistent with the hypothesis that *daf-16* works through *lin-41* to block *col-19p::gfp* expression during dauer (Fig 3B). To ensure that the observed suppression was not due to the *lin-41(xe8)* strain being less sensitive to RNAi, we tested the response of *lin-41(xe8)* and control strains to *unc-22* RNAi and found a similar response in both strains (S8 Fig).

### *lin-29* is not required for *col-19p::gfp* expression in *lin-41(-)* dauer larvae

During continuous development, *lin-41* opposes *col-19p::gfp* and adult cell fate by directly repressing the translation of the LIN-29 transcription factor [30,32]. LIN-29 is in turn a direct activator of *col-19* transcription and is thought to be the most downstream regulator of other aspects of adult cell fate [26,27,31]. If the *col-19p::gfp* expression observed in *lin-41(RNAi)* dauer larvae is due to misexpression of LIN-29, then loss of *lin-29* should prevent *col-19p::gfp* expression. To ask this question, we used the *lin-29(xe37)* deletion allele that removes all but 27 amino acids of LIN-29 [45]. Surprisingly, we saw no effect of the loss of *lin-29* on the expression of *col-19p::gfp* in *lin-41(RNAi)* dauer larvae (Fig 4A). As a control, we established that our strain produced the expected *lin-29(0)* phenotypes in adults: a drastic reduction of *col-19p::gfp* and a complete lack of adult alae formation (S9 Fig).

We next confirmed the *lin-41* RNAi results using the null allele *lin-41(n2914)*. Since *lin-41 (0)* hermaphrodites are sterile [30], we used an available transgene that is integrated close to

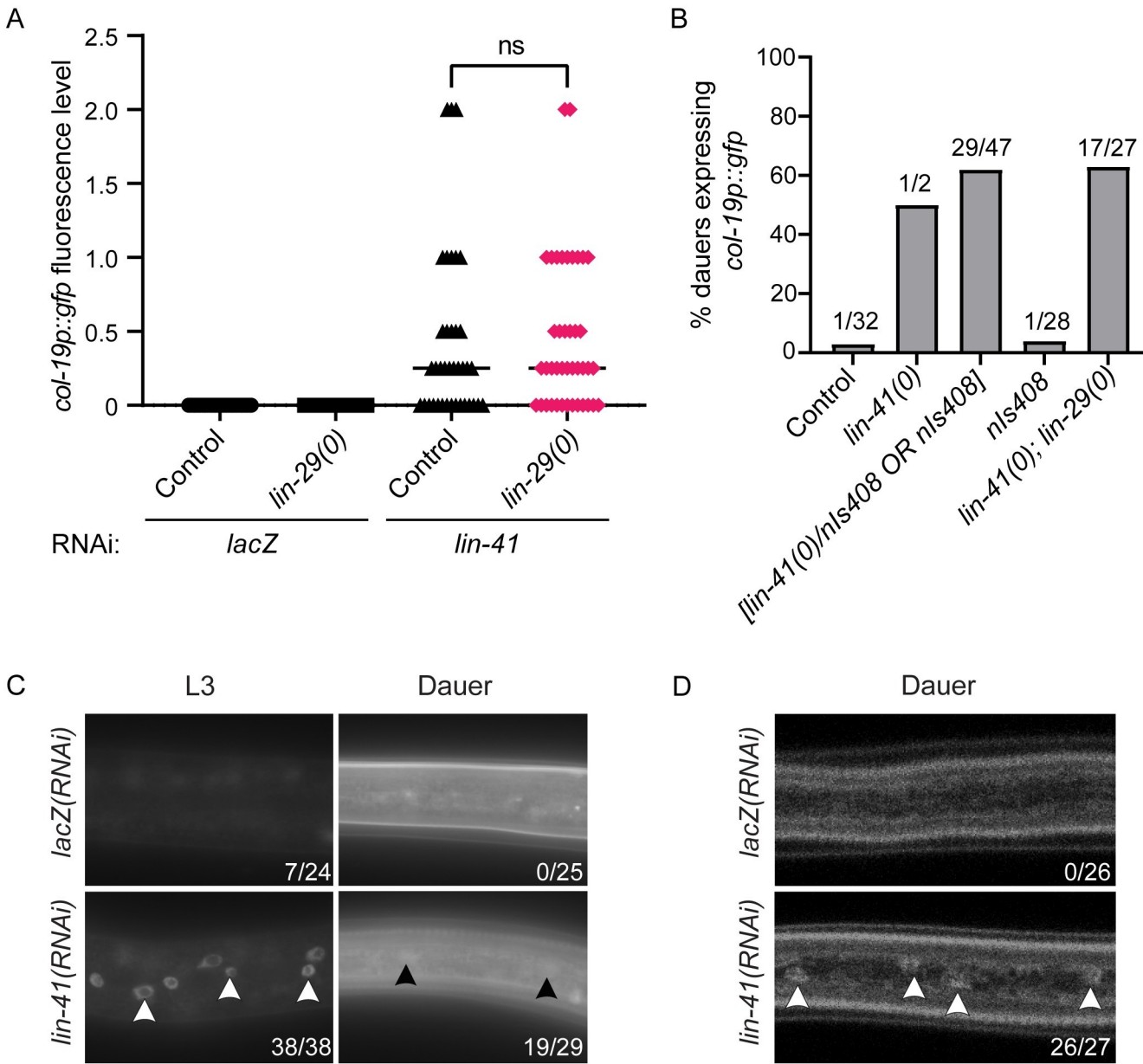

**Fig 4. *lin-41* regulates *col-19p::gfp* independently of *lin-29* in dauer larvae.** (A) *lin-29* is not required for *col-19p::gfp* expression in *lin-41(RNAi)* dauers. Fluorescence levels were determined as described in S7 Fig. ns = not significant (p = 0.4856, two-tailed Mann-Whitney Test). n = 33–41; see S7 Table for complete underlying numerical data. (B) *lin-41(0); lin-29(0)* dauer larvae expressed *col-19p::gfp*. *nIs408* is a transgene used to balance *lin-41(0)* (see text). *lin-41(0)* larvae are dauer-defective [47]. Numbers indicate the number of dauers expressing *col-19p::gfp* over total dauers scored. (C, D) *lin-29::gfp* expression imaged on a compound microscope (C) or a confocal microscope (D). Images were taken with a 63x objective. Arrowheads indicate hypodermal nuclei with visible *lin-29::gfp* expression. Numbers indicate the number of larvae expressing any detectable *lin-29::gfp* over total number of larvae. *lin-41(RNAi)* produced substantial precocious expression of *lin-29::gfp* in continuously developing L3 staged larvae, easily discernable under the compound microscope. However, *lin-41(RNAi)* produced only very dim expression in dauer larvae. Dauer larvae produce substantial autofluorescence at the microscope settings required to visualize *lin-29::gfp*.

the *lin-41* locus as a balancer: *nIs408[lin-29::mCherry, ttx-3p::gfp]* [46]. This transgene had the added advantage of rescuing some of the *lin-29(0)* defects, making the strain easier to maintain (see Methods). When we induced dauer formation in the progeny of *lin-41(0)/nIs408* and *lin-*

*41(0)/nIs408; lin-29(0)* mothers, we found that *lin-41(0); lin-29(+)* homozygous larvae, recognized by the lack of *ttx-3p::gfp* expression, were largely dauer defective (see [47]). Of the two *lin-41(0)* dauer larvae we recovered, one expressed *col-19p::gfp* (Fig 4B). Interestingly, 62% of larvae expressing *ttx-3p::gfp* displayed *col-19p::gfp* expression. This percentage is close to the 2/3 of transgene-containing larvae segregating from the heterozygous parent that would be predicted to be *lin-41(0)/nIs408* heterozygotes, suggesting that *lin-41* is haploinsufficient in its role in blocking *col-19p::gfp* during dauer. To bolster this supposition, we confirmed that dauer larvae homozygous for *nIs408* do not express *col-19p::gfp* (Fig 4B).

Although *lin-41(0)* homozygous larvae display a dauer-defective phenotype, *lin-41(0); lin-29(0)* larvae do not [47]. We were therefore able to test *lin-41(0); lin-29(0)* dauer larvae for *col-19p::gfp* expression. Approximately half of these larvae expressed *col-19p::gfp* (Fig 4B). Although the dauer-defective phenotype of *lin-41(0)* single mutants prevented us from determining the extent to which loss of *lin-29* suppresses the *col-19p::gfp* phenotype in *lin-41(0)* homozygous mutant dauer larvae, these results do confirm that *col-19p::gfp* can be misexpressed in *lin-41(0)* dauer larvae even in the absence of *lin-29*.

To better understand the relationship between *lin-41* and *lin-29* during dauer, we asked whether *lin-41* is required to block LIN-29 expression in dauer larvae. During continuous development, *lin-41* mutants display precocious hypodermal expression of LIN-29 [30,32]. We used *lin-29(xe61[lin-29::gfp])*, which tags both isoforms of *lin-29* [32] to assess the effect of *lin-41* RNAi during dauer. Using settings that allow unambiguous visualization of hypodermal *lin-29::gfp* during L4 and adult stages, we saw only extremely dim *lin-29::gfp* expression in *lin-41(RNAi)* dauer larvae, just at the edge of detection (Fig 4C). We used confocal microscopy as a more sensitive assay to confirm that *lin-29::gfp* is expressed, albeit at very low levels in *lin-41(RNAi)* but not control dauer larvae (Fig 4D). In control RNAi experiments run in parallel, *lin-41(RNAi)* was able to cause strong *col-19p::gfp* expression in dauer larvae (24/27 larvae examined), demonstrating that there were no technical problems in the RNAi experiment that would lead to such low expression. Finally, we confirmed that *lin-41(RNAi)* produces the expected precocious *lin-29::gfp* expression in L3 staged larvae that developed continuously (Fig 4C). Taken together, contrary to the role of *lin-29* during continuous development, our data indicate that *lin-41* regulates *col-19p::gfp* expression in dauer larvae and that *lin-29* does not play a significant role in this regulation.

### *daf-16* acts at least partially independently of *lin-29* to regulate *col-19p::gfp* during dauer

As described above, *daf-16* promotes *lin-41* expression during dauer, and misexpression of *lin-41* suppresses the precocious *col-19p::gfp* expression observed in *daf-16(0)* dauer larvae. Since *lin-29* was not required for misexpression of *col-19p::gfp* in *lin-41(-)* dauer larvae, we hypothesized that *lin-29* would also be dispensable for *col-19p::gfp* expression in *daf-16(-)* dauer larvae. We found that neither of two *lin-29* null alleles affected the penetrance of *col-19p::gfp* expression in *daf-16(0)* mutant dauers, demonstrating that the misexpression of *col-19p::gfp* in *daf-16(0)* dauer larvae does not depend on *lin-29* (Fig 5A). However, when we compared levels of expression between the strains, we found a small but statistically significant decrease in expression in dauer larvae that lack *lin-29*, indicating that the presence of *lin-29* bolsters *col-19p::gfp* expression slightly (Fig 5B). We next asked whether loss of *daf-16* affects expression of *lin-29*. Examining our mRNA-seq data, no significant difference in *lin-29* mRNA levels were observed in *daf-16(0)* vs. control dauer larvae (S6 Fig). However, during continuous development, regulation of *lin-29a* by *lin-41* occurs translationally and may not be evident from mRNA levels [32,48]. We next examined *lin-29::gfp* expression in *daf-16(0)* dauer larvae. Unlike the dim *lin-*

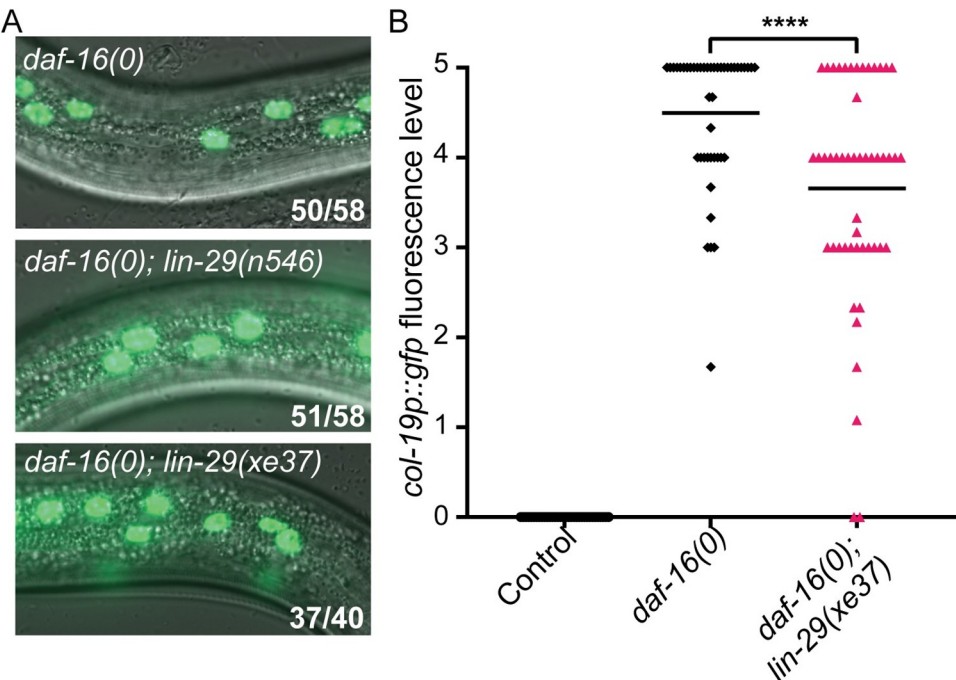

**Fig 5. *daf-16* acts at least partially independently of *lin-29* to block adult cell fate during dauer.** (A) *daf-16(0); lin-29(0)* dauers expressed *col-19p::gfp* at a similar penetrance to *daf-16(0)* dauers. *n546* is a nonsense mutation and *xe37* is a deletion of all but 27 amino acids. Images were taken with a 63x objective. Numbers indicate the number of dauers expressing hypodermal *col-19p::gfp* over the total number of dauers. (B) Levels of *col-19p::gfp* are slightly reduced in *daf-16(0); lin-29(xe37)* dauer larvae compared to *daf-16(0)* dauers. Fluorescence levels were determined as described in S7 Fig. ****p<0.0001 (Two-tailed Mann-Whitney Test). n = 34–50; see S8 Table for complete underlying numerical data.

*29::gfp* expression we observed in *lin-41(RNAi)* dauer larvae, *lin-29::gfp* expression in the hypodermis was completely undetectable in *daf-16(0)* dauer larvae. Using confocal microscopy, 0/24 *daf-16(0)* dauer larvae displayed detectable *lin-29::gfp*. Taken together, these experiments demonstrate that *daf-16* regulates *col-19p::gfp* expression largely independently of *lin-29*.

## *daf-16*/FOXO regulates the expression of many genes during dauer

Since the key downstream regulator of *col-19* expression is not required for the misexpression of *col-19p::gfp* in *daf-16(0)* dauer larvae, one or more different regulators must activate or derepress *col-19p::gfp* expression in this context. One possibility is that DAF-16 itself is a direct activator. Two pieces of indirect evidence initially argued against this possibility. First, the promoter sequence used in the *col-19p::gfp* transgene does not contain a canonical DAF-16/FOXO binding element (TGTTTAC) (S10 Fig) [49]. Furthermore, ChIP-seq experiments performed as part of modENCODE and displayed on WormBase did not show binding of DAF-16 to the *col-19* promoter [50]. However, this evidence is not conclusive. The consensus sequence for DAF-16/FOXO binding elements is very broad, leaving open the possibility that DAF-16 could recognize a non-canonical site [51]. In addition, the ChIP-seq experiments were carried out during the L4/young adult stage, whereas the only time that we see aberrant *col-19p::gfp* expression in *daf-16(0)* mutant larvae is during dauer. To more definitively test DAF-16 binding at the *col-19* promoter during the dauer stage, we used an endogenously tagged *daf-16::zf1::wrmScarlet::3xFLAG* strain to perform ChIP-qPCR during dauer. As expected, the promoter for a confirmed DAF-16 target, *mtl-1* [52,53] showed 18–30 fold

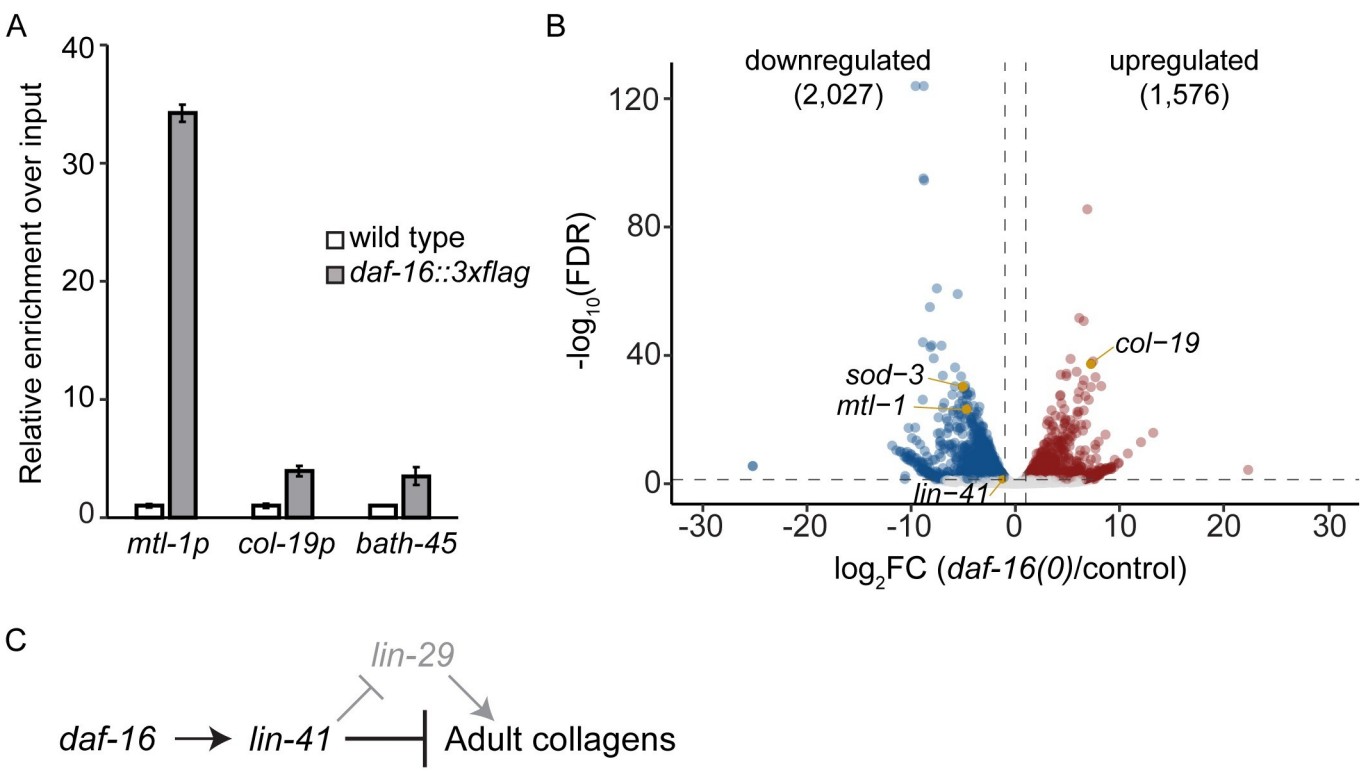

**Fig 6. Gene expression changes in *daf-16(0)* dauer larvae.** (A) DAF-16 is not enriched at the *col-19* promoter. ChIP-qPCR experiments were performed on N2 or *daf-16(ar620[daf-16::zf1-wrmScarlet-3xFLAG])* dauer larvae. Binding of DAF-16-3xFLAG was first normalized to input, and then to the average of the respective wild-type value (mean +/- SD for two technical replicates) is shown. Binding to the *col-19* promoter was not observed, whereas there was substantial binding to the promoter of the known DAF-16 target *mtl-1* [52,53]. The coding region of the heterochromatinized gene *bath-45* was used as negative control. A second biological replicate is shown in S11 Fig, and raw data are shown in S9 Table. (B) Volcano plot showing gene expression changes observed in *daf-16(0)* dauer larvae compared to control dauer larvae. Full mRNA-seq data can be found in NCBI under GEO accession number GSE179166. *mtl-1* and *sod-3* are known transcriptional targets of DAF-16 [52,53,74,75]. (C) Genetic model of the regulation of *col-19p::gfp* and adult cell fate during dauer, based on data presented in the text.

enrichment in the *daf-16::3xflag* ChIP sample compared to wild type. By contrast, the promoter for *col-19* showed only 2-4-fold enrichment in the *daf-16::3xflag* ChIP sample compared to wild type (Figs 6A and S11). This lower level of DAF-16 enrichment at the *col-19* promoter was comparable to the DAF-16 enrichment at the coding region of the heterochromatinized gene *bath-45*, suggesting that the DAF-16 ChIP-qPCR experiments were inherently noisy. These data indicate that *daf-16* blocks *col-19p::gfp* expression indirectly.

To identify candidate regulators of *col-19p::gfp* and endogenous adult collagen expression during dauer, we looked more broadly at the changes in gene expression that occur downstream of *daf-16*/FOXO using our mRNA-seq data comparing *daf-16(0)* vs. control dauer larvae. We found that 2027 genes were downregulated ≥2-fold in *daf-16(0)* dauers, and 1576 genes were upregulated ≥2-fold in *daf-16(0)* dauers (FDR ≤ 0.05) (Fig 6B). Interestingly, when we performed functional annotation clustering on these differentially expressed genes using the Database for Annotation, Visualization, and Integrated Discovery (http://david.abcc.ncifcrf.gov/), terms related to collagens and cuticle structure were highly enriched among both downregulated and upregulated genes (S12 and S13 Figs). Signaling-related terms were also highly enriched, including terms related to ion transport associated with the downregulated genes and terms related to protein kinases and phosphatases associated with the upregulated genes (S12 and S13 Figs). Of the genes whose expression was affected, 112 encode

transcription factors (91 downregulated and 21 upregulated) and are candidate genes to directly regulate *col-19p::gfp* expression and adult cell fate during dauer (S14 Fig).

## Discussion

Heterochronic genes specify stage-specific seam cell fate at each larval stage and at adulthood [24]. During continuous development, heterochronic genes function as a cascade where successive microRNAs act as molecular switches to downregulate early cell fate-promoting transcription factors and RNA-binding proteins, allowing progression to the next cell fate [25,54]. The decision to enter dauer drastically alters the timing of developmental progression [2]. Perhaps for this reason, extensive modulation of the heterochronic pathway has been identified in pre- and post-dauer stages [3,55–57]. For example, many heterochronic genes that are essential for stage-specific cell fate specification during continuous development are dispensable after dauer [3,42,43,55]. However, the mechanisms that act to prevent precocious specification within the dauer stage itself have not been addressed. Here, we find a novel role for *daf-16/* FOXO as a dauer-specific heterochronic gene. *daf-16* opposes adult cell fate during dauer via a modified heterochronic pathway involving *lin-41* but not *lin-29*.

The heterochronic gene *lin-41* encodes an RNA-binding protein that promotes larval cell fate and opposes adult cell fate during continuous development [30,58]. We found that *lin-41* acts downstream of *daf-16* to block adult cell fate during dauer. Specifically, both *daf-16* and *lin-41* are required to prevent precocious expression of the *col-19p::gfp* adult cell fate marker during dauer, and *lin-41* expression is downregulated in *daf-16(0)* dauer larvae (Figs 1A, 2B and 3A). In contrast, the other early-promoting heterochronic genes tested, *lin-14* and *hbl-1*, do not appear to play a role in regulating *col-19p::gfp* hypodermal expression during dauer, and their expression was not significantly affected in *daf-16(0)* dauer larvae (Figs 2B and S6). Finally, a *lin-41* gain-of-function allele partially blocked the misexpression of *col-19p::gfp* caused by *daf-16* RNAi (Fig 3B). All together, these data are consistent with a linear pathway whereby *daf-16* positively regulates *lin-41* during dauer to oppose *col-19p::gfp* and adult cell fate (Fig 6D). These findings do not exclude the possibility that *daf-16* also regulates *col-19p:: gfp* and adult cell fate in parallel to *lin-41*.

The mechanism by which *daf-16* promotes *lin-41* expression is currently unknown. One possibility is that *daf-16* regulates expression of *lin-41* via the *let-7* microRNA. During continuous development, *lin-41* is expressed in early stages but is downregulated by the *let-7* microRNA during the L4 stage to allow progression to adult cell fate. *let-7* opposes *lin-41* expression by binding to the *lin-41* 3'UTR and mediating silencing [30,58]. The ability of the *lin-41(xe8 [Δ3'UTR])* allele to interfere with the *daf-16(-)* phenotype suggests the possibility that *let-7* is involved in this regulatory pathway.

During continuous development *lin-41* blocks adult cell fate by directly repressing the translation of the most downstream component in the heterochronic pathway, the LIN-29 transcription factor that promotes all aspects of adult cell fate [24,26,27,31,32]. LIN-29 expression remains low in the lateral hypodermis during early larval development due to the combined action of *lin-41* and *hbl-1* which regulate distinct *lin-29* isoforms [30–32]. Surprisingly, we found that *daf-16* and *lin-41* regulate adult cell fate mostly independently of *lin-29* (Fig 6D). During continuous development, the precocious phenotypes observed in *lin-41(-)* larvae are completely suppressed by compromising *lin-29* activity [30]. Indeed, the phenotypes of every precocious heterochronic mutant tested during continuous development are suppressed by *lin-29(-)* alleles [26,30,42,43,59]. In contrast, loss of *lin-29* had little to no effect on the precocious *col-19p::gfp* phenotype observed in *daf-16(-)* or *lin-41(-)* dauer larvae (Figs 4A, 4B, 5A and 5B). Therefore, whereas *lin-29* is a direct transcriptional activator of *col-19* and other

adult-enriched collagens in the context of adults [27–29], our work demonstrates that during dauer, *daf-16* blocks adult collagen expression through factor(s) other than *lin-29*. We found over 3600 genes whose expression changed in *daf-16(0)* dauer larvae, including 112 transcription factors. One or more of these factors may control the expression of the *col-19p::gfp* transcriptional reporter.

In addition to regulating *col-19p::gfp*, we found that *daf-16* is required during dauer to block expression of nearly all adult-enriched collagens. The expression of adult-enriched collagens in *daf-16(0)* dauer larvae is accompanied by a concomitant decrease in the expression of dauer-enriched collagens (Fig 1E). The effect of *daf-16* mutations on the dauer cuticle may explain some of the previously observed defects in *daf-16(0)* dauer larvae. Two defining features of dauer larvae are the presence of dauer alae on the cuticle and resistance to treatment with detergents such as SDS. SDS-resistance depends in part on the specialized dauer cuticle [2]. *daf-16(0)* dauer larvae possess dauer alae that are slightly less defined than wild-type, and *daf-16(0)* dauers are only partially SDS-resistant [15,34,60,61]. It is possible that the shift from dauer-enriched collagens to adult-enriched collagens in *daf-16(0)* dauer larvae is responsible for these phenotypes. Notably, our functional annotation clustering analysis found that collagen-related terms were highly enriched among both upregulated and downregulated genes. This finding suggests that regulation of stage-specific collagen expression is a key role of *daf-16*/FOXO during dauer.

Dauer interrupts development midway through the larval stages [2]. During dauer, progenitor cells that have not yet completed development must maintain or re-establish multipotent fate, neither differentiating prematurely nor losing their previously acquired tissue identity. The role we identified for *daf-16* in blocking adult cell fate in lateral hypodermal cells during dauer appears to contribute to the maintenance of multipotency in lateral hypodermal seam cells. We have previously described an analogous role for *daf-16* in promoting multipotent VPC fate [5]. In both contexts, *daf-16* activity is important to prevent precocious adoption of cell fates that normally occur later in development, after recovery from dauer. However, the mechanisms by which these cell fates are adopted differ in each cell type. Adult seam cell fate is regulated by the heterochronic genes, whereas VPC fate specification is regulated primarily by EGFR/Ras and LIN-12/Notch signaling [24,25,62]. The ability of *daf-16*/FOXO to influence these distinct developmental pathways suggests that *daf-16*/FOXO has a broad role in promoting multipotent cell fate during dauer. As one of the major regulators of dauer formation, *daf-16* is well-positioned to coordinate the decision to enter dauer with the necessary alterations to developmental pathways that are paused during dauer.

*daf-16* is the sole *C. elegans* ortholog of the genes encoding the FOXO proteins [15,16]. FOXO transcription factors regulate both stem cell quiescence and stem cell plasticity across species, from *Hydra* to mammals [20,63]. However, the mechanisms by which FOXO promotes multipotency in stem cells are still emerging. The protein encoded by the *lin-41* ortholog, LIN41/TRIM71 also promotes cell fate plasticity in mammalian stem cells [64]. Our work provides the first connection between *daf-16*/FOXO and *lin-41*/TRIM71 and may be relevant to mammalian stem cells.

## Methods

### Strains and maintenance

A full list of strains and their genotypes used in this study is located in S1 Table. Balanced strains are described in more detail here. Homozygous *lin-41(0)* hermaphrodites are sterile, therefore for experiments involving the null allele *lin-41(n2914)*, *lin-41* was balanced with transgene *nIs408[ttx-3p::gfp, lin-29::mCherry]* [46], which we found to be closely linked to *lin-41*. At each generation, larvae that expressed *ttx-3p::gfp* were singled out and their progeny

were monitored for segregation of *lin-41(0)* homozygotes that were Dpy, Ste, and lacked *ttx-3p::gfp*. The rescuing *lin-29::mCherry* enabled more robust growth in the *lin-41(0)/nIs408; lin-29(0)* strain than typical strains homozygous for *lin-29(0)*. Although *lin-29(0)* mutants are homozygous viable, they are Egl, Pvl, and have reduced brood size [24]. For experiments, *lin-41(n2914)* homozygous larvae were identified based on the lack of *ttx-3p::gfp* expression. For more details about experiments involving these strains see Cale & Karp (2020) [47].

Animals homozygous for the *lin-41(xe8[Δ3'UTR])* allele burst at young adulthood [44]. This gain-of-function allele was balanced over the *lin-41(bch28 xe70)* allele, where *bch28* is a complex insertion of *eft-3p::gfp* that disrupts the *lin-41* locus, and *xe70* is a deletion of the *lin-41* 3'UTR [45,65]. For experiments, *lin-41(xe8)* larvae were identified based on the lack of *eft-3p::gfp* expression.

All strains were grown according to standard procedures on Nematode Growth Medium (NGM) plates seeded with the *E. coli* strain OP50 [66]. Strains were maintained at 15˚C or 20˚C.

## Dauer induction

All strains used for experiments involving dauer larvae contained *daf-7(e1372)*, which is a temperature-sensitive, hypomorphic allele that induces dauer entry at 24˚C or 25˚C [34,67]. Unless otherwise specified, dauer larvae were obtained by allowing 10–20 gravid adult hermaphrodites to lay embryos for 2–8 hours at 24˚C, then removing the parents and allowing the embryos to incubate at 24˚C. For all experiments except those in Fig 1B, dauer larvae were scored approximately 48–52 hours after egg-laying, a time soon after dauer formation ("early dauer" or 0-day dauer larvae). In Fig 1B when different stages were examined, L2d larvae were scored at 24 hours after egg-laying; L2d-dauer molt larvae were scored at 39 hours after egg-laying; 1-day dauer larvae were scored at 72 hours after egg-laying. Dauer formation was verified by looking for crisp, defined dauer alae and radial constriction [67].

For experiments involving homozygous egg-laying-defective strains, including those with *lin-29(0)* or *lin-29(xe61)*, embryos were obtained by sodium hypochlorite treatment consisting of two 2-minute incubations in 1M NaOH, 10% Clorox bleach. For experiments with XV254 *daf-16(mgDf50); lin-29(xe37); daf-7(e1372); maIs105[col-19p::gfp]* and controls, embryos were obtained by dissecting gravid adult hermaphrodites.

## Nondauer stages

**Continuous development (Fig 1C).** Strains XV33 *maIs105[col-19p::gfp]* and VT1750 *daf-16(mgDf50); maIs105* were synchronized by allowing gravid adult hermaphrodites to lay embryos at 24˚C or 25˚C and then incubating the progeny until the desired stage was reached. Developmental stage was evaluated by the extent of gonad and/or vulval development and scored for *col-19p::gfp* or *lin-29::gfp* expression.

**L1 arrest (Fig 1C).** To induce L1 arrest, embryos from strains XV33 and VT1750 were obtained by sodium hypochlorite treatment and then incubated at 20˚C in M9 in a shaking incubator. L1 larvae were removed each day for 7 days and scored for *col-19p::gfp* expression.

**Adults (S9 Fig).** Strains VT1777 *daf-7(e1372); maIs105* and XV253 *lin-29(xe37); daf-7(e1372); maIs105* were synchronized by sodium hypochlorite treatment. Embryos were placed on seeded NGM plates and grown at 20˚C for 72–75 hours. Adults were scored for the presence of stage-specific alae and *col-19p::gfp* expression.

## RNAi

RNAi bacteria were from the Ahringer library (Source Bioscience), except the *lacZ* RNAi clone was pXK10 [68]. RNAi plates were prepared by adding 300μL of an overnight culture of RNAi

bacteria to 60mm NGM plates containing 50μg/mL carbenicillin and 200μg/mL IPTG. Embryos obtained from sodium hypochlorite treatment were plated onto seeded RNAi plates and incubated at 24°C to allow *daf-7(e1372)* larvae to enter dauer or *daf-7(+)* larvae to develop continuously.

### Compound microscopy

Animals were picked onto slides made with 2% agarose pads and paralyzed with 0.1 M levamisole. A Zeiss AxioImager D2 compound microscope with HPC 200 C fluorescent optics was used to image worms. DIC and fluorescence images were obtained using a AxioCam MRm Rev 3 camera and ZEN 3.2 software. GFP was visualized with a high efficiency GFP shift free filter.

### Confocal microscopy

Animals were picked onto slides made with 2% agarose pads and paralyzed with 0.1 M levamisole. A Nikon A1R scanning laser confocal light microscope was used to image worms using an excitation laser set to at wavelength of 488 nm. To visualize *lin-29*::*gfp*, the laser power was set to 70%.

### Phenotypic data collection and analysis

All phenotypic data presented are from at least two independent experiments, typically performed by independent researchers. Any experiments that involved subjective decisions were blinded by having a lab member not involved in the experiment code the strains and/or images before scoring. Statistical analyses were performed on Graphpad Prism (version 9.1) and specific tests are described in individual figure legends. P-values < 0.05 were deemed statistically significant.

### Sample collection for qPCR and RNA sequencing

Synchronized populations of VT2317 *daf-16(mgDf50); daf-7(e1372)* and control CB1372 *daf-7 (e1372)* dauer larvae were obtained by incubating embryos isolated by sodium-hypochlorite treatment at 24°C for 52 hours. Because some *daf-16; daf-7* animals grown under these conditions fail to enter dauer [60], dauer larvae from both strains were handpicked into M9 solution, washed twice, and then pelleted to a volume of 100μl packed worms. TRIzol reagent (Invitrogen) was added to the samples at a 10:1 ratio of TRIzol to worm pellet. The samples were then frozen in dry-ice/ethanol. Two biological replicates were obtained for each strain for mRNA-seq. For qPCR, we obtained two biological replicates of the wild-type sample and three biological replicates of the *daf-16(0)* mutant sample.

### RNA isolation

Total RNA isolation from dauer sample was conducted using TriReagent (Ambion) protocol with the following modifications: pelleted *C. elegans* in TRI-Reagent were subjected to three freeze/thaw/vortex cycles prior to BCP addition to improve extraction efficiency, isopropanol precipitation was conducted in the presence of glycogen for 1hr at -80°C, RNA was pelleted by centrifugation at 4°C for 30 minutes at 20,000 x g; the pellet was washed three times in 70% ethanol and resuspended in water. BioAnalyzer assay (Agilent Technologies) was used for quality control of the RNA sequencing samples prior to library creation, with a minimum RIN of 8.5. NanoDrop2000 (Thermo Scientific) was used to quantify and assess the RNA quality of samples analyzed by qPCR.

## RT-qPCR

cDNA was synthesized from 250ng total RNA using SuperScript III Reverse Transcriptase (Invitrogen) and analyzed with a CFX96 Real-Time System (BioRad) using Absolute Blue SYBR Green PCR MasterMix (Life Technologies). Relative *lin-41* mRNA levels were calculated based on the ΔΔ2Ct method [69] using *eft-2* for normalization. Results presented are the average values of independent calculations from biological replicates.

**RT-qPCR primers.**

| | |
|---|---|
| *eft-2* F | ACGCTCGTGATGAGTTCAAG |
| *eft-2* R | ATTTGGTCCAGTTCCGTCTG |
| *lin-41* F | GGTTCCAAATGCCACAAGAG |
| *lin-41* R | AGGTCCAACTGCCAAATCAG |

## mRNA-seq library preparation and sequencing

Libraries were constructed using the TruSeq RNA Library Prep Kit v2 (Illumina). The DNA concentration and fragment size of sequencing libraries were analyzed using the BioAnalyzer assay (Agilent Technologies). High-throughput sequencing was performed on the Illumina HiSeq 2500 platform to generate paired-end reads of 100 bp.

## mRNA-seq analysis

Basecalling and base call quality were performed using Illumina's Real-Time Analysis (RTA) software, and CIDRSeqSuite 7.1.0 was used to convert compressed bcl files into compressed fastq files. Using Trimmomatic v. 0.39, Illumina adapters were clipped from raw mRNA-seq reads, followed by quality trimming (LEADING: 5; TRAILING: 5; SLIDINGWINDOW:4:15) [70]. Reads with a minimum length of 36 bases were retained (MINLEN: 36). Processed reads were aligned to *C. elegans* reference genome WBcel235 using STAR v. 2.4.2a with default parameters and—twopassMode Basic [71]. RSEM v. 2.1 was used to quantify gene abundance based on mapped reads [72]. Outlier samples were identified by assessing the quality of raw and processed reads with FastQC v. 0.11.7 (http://www.bioinformatics.babraham.ac.uk/projects/fastqc/) and by Principal Component Analysis (PCA) on $\log_2$-transformed normalized gene expression counts generated using DESeq2 v. 1.30.0 [73]. Differential gene expression analysis was performed using DESeq2 with a significance threshold of FDR $\leq$ 0.05 and absolute value of $\log_2$-transformed fold changes of at least 2.

## DAVID analysis

Differentially upregulated and downregulated genes with at least a $\log_2$-transformed fold change of 2 and FDR $\leq$ 0.05 were individually analyzed using DAVID Functional Annotation Clustering (https://david.ncifcrf.gov/summary.jsp. Accessed 20210507). Terms from InterPro and Gene Ontology sub-ontologies cellular component (CC), biological process (BP), and molecular function (MF) were plotted by $-\log_{10}$(FDR) and ranked by fold enrichment.

## ChIP-qPCR

GS8924 (*daf-16(ar620[daf-16::zf1-wrmScarlet-3xFLAG]*)), a gift from Katherine Luo and Iva Greenwald (Columbia University) and control N2 embryos obtained after sodium-hypochlorite treatment were hatched overnight in M9 buffer. The L1 arrested worms were grown on NGM plates seeded with HB101 at 25°C. After 5 days, the starved worm population was collected in M9 buffer and nutated in 1% SDS solution for 30 minutes to isolate dauers. The worm pellet was washed 4x with autoclaved DI water and the worms were plated on an NGM

plate with no food. The live dauers crawled out and were collected as a ~500μL packed pellet in M9 buffer.

The animals were nutated for 30min at room temperature in 12mL of 2.6% formaldehyde in autoclaved DI water for live crosslinking. The crosslinking was quenched with 600μL of 2.5M glycine for 5min at room temperature. The worms were washed 3x in water before flash freezing and storing at -80˚C. Frozen pellets were ground twice for 1min in the Retsch MM400 cryomill at a frequency of 30/s. The worm powder was lysed in 2mL of 1xRIPA (1x PBS, 1% NP40, 0.5% sodium deoxycholate, 0.1% SDS) with 1xHALT Protease and phosphatase inhibitor (Thermo Scientific 78443) for 10min at 4˚C. The crosslinked chromatin was sonicated in the Diagenode BioruptorPico for 3 3min cycles, 30s ON/OFF at 4˚C. The chromatin was diluted to a concentration of 20-30ng/uL. 10% of the volume was processed as the input sample. 10ug of chromatin was incubated with 2ug of Monoclonal Anti-Flag M2 antibody (Sigma-Aldrich F3165) overnight at 4˚C and then for 2 hours with Dynabeads M-280 Sheep Anti-Mouse IgG (Invitrogen 11201D). After 3 800uL washes in LiCl buffer (100mM Tris HCl pH 7.5, 500mM LiCl, 1% NP40, 1% sodium deoxycholate), the samples were de-crosslinked by incubating with 80ug of Proteinase K (Thermo Scientific 25530015) in Worm Lysis buffer (100mM Tris HCl pH 7.5, 100mM NaCl, 50mM EDTA, 1% SDS) at 65˚C for 4 hours. The samples were subjected to phenol-chloroform extraction and the DNA pellet was resupended in TE buffer. RNase A (Thermo Scientific 12091021) treatment was performed for 1 hour at 37˚C.

Quantitative PCR for promoter regions of interest was performed with Absolute Blue SYBR Green (Thermo Scientific AB4166B) on the CFX96 Real Time System Thermocyclers (Biorad) using custom primers. The cycle numbers in the ChIP samples were normalized to respective input values. The log 2 transformed fold change values in the *daf-16::3xFLAG::wrmScarlet* samples were normalized to the respective N2 samples. 2 biological replicates with 2 technical replicates each were done for the promoter regions of *mtl-1* and *col-19*. Each biological replicate was analyzed separately.

**ChIP-qPCR primers.**
*mtl-1p* F    TGAGCACTCTAATCCTTTGCAC
*mtl-1p* R    ACGTGAATGTTGCAAACACCT
*col-19p* F   TCCATCTCTCTTGGAAACACAT
*col-19p* R   ACACCTTCAAACCTAACCAGTGT
*bath-45* F   ATTTAATTTGTTTTCAGAAGGCAGC
*bath-45* R   GTGCTTTCTGCATCAGAGCC

## Supporting information

**S1 Fig. *daf-5* is not required to block *col-19::gfp* expression during dauer.** The *daf-7*/TGFβ pathway regulates dauer formation in parallel to the insulin-like pathway [6,7]. *daf-5* encodes a Sno/Ski protein that works in a complex with DAF-3/SMAD to regulate transcription downstream of DAF-7/TGFβ signaling [76]. *daf-5(0)* mutants are dauer-defective, but can be induced to enter dauer when combined with the *daf-2(e1370)* allele [34,35]. (A) *daf-2(e1370)* mutants develop slowly and acquire dauer characteristics approximately one day later than wild-type or *daf-7(e1372)* larvae [60,77]. Since *daf-16; daf-7* dauer larvae express *col-19p*::gfp most penetrantly within one day of dauer entry, we used SDS-resistance to determine a time that might correspond to "early dauer" in this strain. SDS-resistance is acquired at the end of the L2d-to-dauer molt and continues throughout dauer [2]. Gravid adult hermaphrodites of the indicated strains were allowed to lay eggs for 3 hours and then removed. Embryos were incubated at 24˚C for the indicated times and then incubated in 1% SDS for 10 minutes. The

percent of larvae alive after this treatment is shown; data were aggregated from four independent experiments. (B) Percent of dauer larvae expressing *col-19p*::*gfp*. This expression was never observed in either control or *daf-5(0)* dauer larvae at times when many early dauer larvae should have been present. Data shown were aggregated from 3–4 independent experiments.
(TIF)

**S2 Fig. Dauer larvae lacking 1–2 isoforms of *daf-16* display *col-19p::gfp* more dimly and in fewer cells than *daf-16(0)*.** (A) Representative micrographs of dauer larvae scored in Fig 1D. In Fig 1D, any *col-19p*::*gfp* expression in the lateral hypodermis was considered "on", however there are qualitative differences in that expression in the different strains. Dashed lines in the control indicate the boundaries of the dauer larva, as determined by DIC optics. (B) Loss of 1–2 isoforms of *daf-16* results in fewer cells expressing *col-19p*::*gfp* compared to *daf-16(0)* dauer larvae. Quantification of the number of cells expressing *col-19p*::*gfp* along one side of the lateral hypodermis between the pharynx and rectum. Each dot represents a dauer larva. *p = 0.0460 (Two-tailed Mann-Whitney Test). (C) Loss of 1–2 isoforms of *daf-16* results in dimmer expression of *col-19p*::*gfp* throughout the hypodermis compared to the *daf-16(0)* dauer larvae. Fluorescence levels were determined as described in S7 Fig. ****p<0.0001 (Two-tailed Mann-Whitney Test). Larvae shown in panels B-C were from an independent set of experiments from those in Fig 1D. n = 28–31; see S2 Table for complete underlying numerical data.
(TIF)

**S3 Fig. Seam cells divide and remain unfused in *daf-16(0)* dauer larvae.** (A) Micrographs showing apical junctions between seam cells in *daf-7* (control) and *daf-16(0); daf-7 (daf-16(0))* dauer larva, and seam cell divisions in *daf-16(0)* dauer larvae. The worms are oriented with anterior to the left and ventral down. Apical junctions were visualized by *ajm-1*::*gfp* as part of the *wIs78* transgene [42,78]. Junctions were present between all seam cells in both *daf-7* control and *daf-16(0); daf-7* dauer larvae. Several seam cells in the *daf-16(0); daf-7* dauer shown have recently divided. An example of one dividing seam cell is bracketed. The region bounded by junctions is smaller and rounder in the anterior daughter that will fuse with hyp7 (h). The posterior daughter (s) will remain a seam cell. Seam cells that were not dividing are indicated with an arrow. These photos were taken soon after dauer formation ("early dauers"). (B) Percent of dauer larvae with one or more seam cells actively dividing at the time of scoring. Seam cell divisions were identified based on *ajm-1*::*gfp* expression. Cell divisions were only seen in *daf-16(0)* dauer larvae.
(TIF)

**S4 Fig. *daf-16(RNAi)* dauer larvae express *col-19p::gfp* more strongly than *lin-41(RNAi)* dauer larvae.** Larvae assessed for *col-19p*::*gfp* expression in Fig 2 were analyzed for fluorescence intensity using ImageJ. GFP-positive nuclei were outlined using the freehand tool and the mean pixel intensity for all of such nuclei were averaged together for each individual worm. These averages are underestimates for the expression of *col-19p*::*gfp* in *daf-16(RNAi)* dauer larvae because many nuclei had fully saturated pixels. ****p<0.0001 (Two-tailed Mann-Whitney Test). (n = 30).
(TIF)

**S5 Fig. RNAi of *lin-41* and *hbl-1* induces *col-19p::gfp* expression in non-hypodermal tissues.** Numbers indicate larvae expressing *col-19p*::*gfp* in at least one cell in the relevant tissue over the total number of dauer larvae scored. Dauer larvae are oriented with anterior to the left and ventral down. (Top) *lin-41* RNAi induces *col-19p*::*gfp* expression in vulva precursor cells

(VPCs) during dauer at high penetrance. The larva shown expressed *col-19p*::*gfp* in P5.p and P7.p, but overall which VPCs expressed *col-19p*::*gfp* varied from worm to worm and did not correlate with the predicted cell fate they will adopt. The exposure time in the fluorescence channel was 200ms. (Bottom) *hbl-1* RNAi induces *col-19p*::*gfp* expression in occasional ventral neurons during dauer at lower penetrance. As *col-19p*::*gfp* includes a nuclear localization sequence, expression is typically restricted to the nucleus. However, in the larva shown a neuronal process could be seen extending posteriorly to the cell body (arrow). The exposure time in the fluorescence channel was 50ms.
(TIF)

**S6 Fig. Levels of most core heterochronic genes are essentially unaffected by *daf-16* during dauer.** (A) A diagram of the network of heterochronic genes that regulates stage-specific seam cell fate during continuous development. "3 *let-7s*" indicates the *let-7* family members, *mir-48*, *mir-84*, and *mir-241*. *mir-48* is also downregulated by *lin-42* (not depicted) [79]. All protein-coding genes except *lin-41* are indicated in blue. (B) Log$_2$ fold change of the protein-coding heterochronic genes other than *lin-41*, comparing *daf-16(0)* to control dauer larvae. Log$_2$ fold change was calculated by DESeq2 from mRNA-seq data. Dashed lines indicate 2x fold upregulation or downregulation. False discovery rates (FDR) are indicated for each gene.
(TIF)

**S7 Fig. Fluorescence scale used to determine relative fluorescence intensity of *col-19p::gfp*.** Fluorescence images of *col-19p*::*gfp* in dauer larvae were taken from a strain we found to produce a wide range of expression levels, XV160, *daf-7(e1372); maIs105; unk-1(xk6)*. These images were taken using compound microscopy and identical settings, including an exposure time of 125ms. The images were then exported as 8-bit tiffs for ImageJ analysis. We identified six representative images distributed across the range of ImageJ values (0–250). We assigned numbers (0.5–5) to these images that reflect their distribution. A value of 0.25 was used if expression was visible, but less than that in the 0.5 panel. The fluorescence scale values with ImageJ values in parenthesizes were as follows: 0.5 (22), 1 (45), 2 (104), 3 (149), 4 (200), and 5 (250). These six images were then used as a scorecard to compare to experimental images for Figs 3B, 4A, and 5B. For these experiments, a 63x objective was used to take 2–3 images along the length of the dauer larva, using identical settings for all strains being compared. Each image was then subjectively compared to the fluorescence scale and assigned a value 0–5, with 0.5 values being used if an image was between two whole-number images. The values for each worm were averaged together to create an overall score for the worm.
(TIF)

**S8 Fig. The *lin-41(xe8)* strain is not defective for RNAi.** Unc phenotypes produced by *unc-22(RNAi)* were assessed during forward locomotion and binned into categories Mild: occasional twitching that did not interrupt movement. Intermediate: occasional twitching that did interrupt movement. Severe: constant twitching but still capable of forward locomotion. Paralyzed: worms whose twitching was so severe that they were not able to move forward. *unc-22* RNAi experiments were carried out in parallel to the experiments shown in Fig 3B. However, because dauer larvae did not display strong Unc phenotypes, dauer larvae were moved to new RNAi plates at 20˚C and allowed to recover to the post-dauer L4 (PDL4) stage. Numbers indicate the total number of PDL4 larvae assessed.
(TIF)

**S9 Fig. The *lin-29(xe37)* strain displays reiterative phenotypes in adults.** (A-B) The background for all strains was *daf-7(e1372); maIs105[col-19p::gfp]*. (A) *lin-29(xe37)* mutant adults do not have adult alae. Numbers indicate the number of worms with alae defects over the total

number of adults. (B) *lin-29(xe37)* mutant adults express very low levels of *col-19p::gfp*. The two images were taken with identical settings (10ms exposure). Representative images are shown (n = 37–38).
(TIF)

**S10 Fig. DAF-16 is unlikely to bind the *col-19* promoter to regulate transcription.** (A) Genetic region upstream of *mtl-1*, a confirmed transcriptional target of DAF-16 [52,53]. DNA sequence beginning immediately 3' to W02F12.8, the gene upstream of *mtl-1* is shown. This sequence likely contains promoter elements that regulate *mtl-1* expression. A canonical DAF-16-Binding Element (DBE, red) is located 76bp upstream of the *mtl-1* ATG. (B) The 846bp upstream of the *col-19* ATG which comprises the regulatory sequence driving *col-19p::gfp* expression [28]. No canonical DBEs (GTAAACA or TGTTTAC) [49] were found in this sequence. Sequences were taken from WormBase (WS280).
(TIF)

**S11 Fig. DAF-16 is not enriched at the *col-19* promoter.** ChIP-qPCR experiments were performed on N2 or *daf-16(ar620[daf-16::zf1-wrmScarlet-3xFLAG])* dauer larvae. Binding of DAF-16-3xFLAG was first normalized to input, and then to the average of the respective wild-type value (mean +/- SD for two technical replicates) is shown. Binding to the *col-19* promoter was not observed, whereas there was substantial binding to the promoter of the known DAF-16 target *mtl-1* (Barsyte *et al.* 2001; Li *et al.* 2008). The coding region of the heterochromatinized gene *bath-45* was used as negative control. This figure shows the second of two biological replicates; the first replicate is shown in Fig 6A.
(TIF)

**S12 Fig. Functional annotation clustering for genes upregulated in *daf-16* mutants.** DAVID analysis showing gene ontology (GO) and InterPro terms that were significantly enriched (p ≤ 0.05, Bonferroni corrected) in genes whose expression was upregulated ≥2x (FDR ≤ 0.05) in *daf-16(0); daf-7* dauer larvae vs. *daf-7* (control) dauer larvae. GO BP = GO term biological process; GO CC = GO term cellular compartment; GO MF = GO term molecular function, INTERPRO = InterPro protein classification.
(TIF)

**S13 Fig. Functional annotation clustering for genes downregulated in *daf-16* mutants.** DAVID analysis showing gene ontology (GO) and InterPro terms that were significantly enriched (p ≤ 0.05, Bonferroni corrected) in genes whose expression was downregulated ≥2x (FDR ≤ 0.05) in *daf-16(0); daf-7* dauer larvae vs. *daf-7* (control) dauer larvae. GO BP = GO term biological process; GO CC = GO term cellular compartment; GO MF = GO term molecular function, INTERPRO = InterPro protein classification.
(TIF)

**S14 Fig. Transcription factors differentially expressed in *daf-16(0)* dauer larvae compared to control dauer larvae.** Genes encoding transcription factors were defined based on their WormBase annotation (WS279). Although annotated as transcription factors, we noticed that the WormBase list also contained genes encoding RNA-binding proteins [80,81]. We therefore subtracted genes identified as encoding RNA-binding proteins from the transcription factor list. From among these genes, we searched our mRNA-seq data to find those whose expression changed ≥2x (FDR ≤ 0.05) by DESeq2 analysis. This analysis produced 112 genes that fit our criteria. Those genes are shown here as heat maps depicting relative expression of these genes in *daf-16(0); daf-7* dauer larvae or *daf-7* (control) dauer larvae from mRNA-seq data. The row-

normalized reads for each of two biological replicates per strain are depicted.
(TIF)

**S1 Table. List of strains used in this study.**
(DOCX)

**S2 Table. Underlying numerical data for S2C Fig, fluorescence levels of col-19p::gfp in daf-16 isoform mutant dauers.**
(XLSX)

**S3 Table. Collagen expression during development.**
(XLSX)

**S4 Table. Adult-enriched collagen expression in seam cells and hyp7.**
(XLSX)

**S5 Table. Raw lin-41 qPCR data for Fig 3A.**
(XLSX)

**S6 Table. Underlying numerical data for Fig 3B, fluorescence levels of col-19p::gfp in lin-41(xe8) mutant dauers.**
(XLSX)

**S7 Table. Underlying numerical data for Fig 4A, fluorescence levels of col-19p::gfp in lin-41(RNAi); lin-29(0) dauers.**
(XLSX)

**S8 Table. Underlying numerical data for Fig 5B, fluorescence levels of col-19p::gfp in daf-16(0); lin-29(0) dauers.**
(XLSX)

**S9 Table. Raw ChIP-qPCR data for Fig 6A.**
(XLSX)

## Acknowledgments

We are grateful to Brooklynne Watkins for advice about confocal microscopy and to other members of the Schisa lab (Central Michigan University) for helpful discussions. mRNA-seq library preparation, sequencing, and data processing into fastq format were conducted at the Genetic Resources Core Facility, Johns Hopkins Institute of Genetic Medicine, Baltimore, MD. Computational resources were provided by the Maryland Advanced Research Computing Center (MARCC). Many thanks to Katherine Leisan Luo in the Greenwald lab at Columbia University for sharing *daf-16(ar620)* prior to publication. We thank WormBase. Some strains were provided by the *Caenorhabditis* Genetics Center (CGC).

## Author Contributions

**Conceptualization:** Xantha Karp.

**Data curation:** Margaret R. Starostik.

**Formal analysis:** Amelia F. Alessi, Margaret R. Starostik.

**Funding acquisition:** John K. Kim, Xantha Karp.

**Investigation:** Matthew J. Wirick, Allison R. Cale, Isaac T. Smith, Amelia F. Alessi, Margaret R. Starostik, Liberta Cuko, Kyal Lalk, Mikayla N. Schmidt, Benjamin S. Olson, Payton M.

Salomon, Alexis Santos, Axel Schmitter-Sánchez, Himani Galagali, Kevin J. Ranke, Payton A. Wolbert, Macy L. Knoblock.

**Methodology:** Axel Schmitter-Sánchez.

**Project administration:** Xantha Karp.

**Supervision:** John K. Kim, Xantha Karp.

**Visualization:** Matthew J. Wirick, Allison R. Cale, Isaac T. Smith, Amelia F. Alessi, Margaret R. Starostik, Himani Galagali, Xantha Karp.

**Writing – original draft:** Matthew J. Wirick, Allison R. Cale, Amelia F. Alessi, Margaret R. Starostik, Himani Galagali, Xantha Karp.

**Writing – review & editing:** Matthew J. Wirick, Allison R. Cale, Amelia F. Alessi, John K. Kim, Xantha Karp.

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
