## [Decision Letter · Decision Letter 0]

23 Aug 2021

Dear Dr Karp,

Thank you very much for submitting your Research Article entitled 'daf-16/FOXO blocks adult cell fate in Caenorhabditis elegans dauer larvae via lin-41/TRIM71' to PLOS Genetics.

The manuscript was fully evaluated at the editorial level and by independent peer reviewers. The reviewers appreciated the attention to an important topic but identified some concerns that we ask you address in a revised manuscript

We therefore ask you to modify the manuscript according to the review recommendations. Your revisions should address the specific points made by each reviewer.

[LINK]

Yours sincerely,

Coleen T. Murphy

Associate Editor

PLOS Genetics

Gregory P. Copenhaver

Editor-in-Chief

PLOS Genetics

The reviewers find the work to be interesting and based on solid experimental results, and only request a few modifications.

Reviewer's Responses to Questions

**Comments to the Authors:**

Reviewer #1: Wiric et al present novel and impactful findings about roles for the evolutionarily conserved transcription factor DAF-16/FOXO and the evolutionarily conserved RNA binding protein LIN-41/TRIM71 in maintaining multipotency during developmental arrest in C. elegans dauer larvae. Upon sensing conditions adverse for reproduction, worm second stage larvae can elect to follow an alternative developmental trajectory – the so-called ‘L2D’ followed by formation of the morphologically specialized and developmentally arrested dauer larva. Dauer larvae are non-feeding, stress resistant, and can maintain a state of developmental arrest and full developmental multipotency for months. Upon encountering favorable conditions, dauer larvae resume development and complete ‘post-dauer L3’ and ‘post-dauer L4’ stages, followed by the reproductively competent adult. The dauer larva developmental trajectory (consisting of the L2d, dauer, and post-dauer L3-L4-Adult) has been a fascinating subject of investigation for decades, and genetic and molecular studies of this pathway have revealed many fundamental genes, mechanisms, and principles underlying metabolic and environmental signaling, stress resistance, longevity, and the regulation of developmental progression. This Wiric et al manuscript reports findings that reveal fundamental novel understanding of one of the most compelling properties of dauer larvae, which is the maintenance of stem cell multipotency during extended developmental arrest. The management of stem cell developmental potential during periods of quiescence is a fundamental property of all multicellular systems and is also a phenomenon that is poorly understood. This manuscript represents a major advance in the field, by identifying a gene regulatory pathway in which DAF-16/FOXO functions to maintain expression of the RNA binding protein LIN-41/TRIM71 in dauer larvae, and LIN-41 functions to prevent the inappropriate expression of adult fates during developmental arrest. The authors discovered that daf-16 loss-of-function (lf) dauer larvae dramatically up regulate numerous genes that are ordinarily adult-specific, including whole sets of adult-specific collagens. This finding places daf-16, for the first time, within the heterochronic pathway, which is a gene regulatory network (previously known to include lin-41) that controls the timing of diverse stage-specific cell fate progression events during larval development. Interestingly, the role for daf-16 in the heterochronic pathway is restricted to the dauer life history trajectory; daf-16(lf) has no detectable effect on adult cell fate timing during continuous (non-dauer) development. The author’s epistasis data showing the daf-16 functions upstream of lin-41, which functions in both continuous development and dauer development to prevent precocious expression of adult fates. However, Wiric et al show that lin-41 prevents adult gene expression in dauer larvae independently of the transcription factor, LIN-29, which is essential for adult-specific gene expression during non-dauer development. The author’s transcriptional profiling of daf-16(lf) dauer larvae reveals that daf-16 controls the expression of thousands of genes, in addition to adult-specific collagens, including many transcription factors that could substitute for lin-29 dauer-specifically. These Wiric et al findings, by identifying a daf-16-lin-41 axis for maintaining multipotency in dauer larvae, significantly expand our understanding of the rewiring of developmental gene regulatory networks that occurs in association with development arrest in this system. I particular the authors show that lin-41 represents a major node of this rewiring, wherein daf-16 functions dauer-specifically to promote lin-41 activity (and to thereby maintain larval developmental potency) and lin-29 functions downstream of lin-41 specifically during non-dauer development. The evolutionary conservation of DAF-16/FOXO and LIN-41/TRIM71, and the apparent conservation of their functions in stem cell fate and developmental progression, underscores the impact of these Wiric et al findings to the broader understanding of developmental and tissue homeostatic mechanisms in animals.

I did not note any major issues with the manuscript. I think that it is suitable for publication in PloS Genetics in its present form.

I have just one minor critique: The pathway model shown in Figure 2A shows a positive role for lin-14 in promoting lin-28 activity and shows the let-7family ‘sister’ microRNAs as regulating only hbl-1. However, an alternative depiction of the network would place the let-7family microRNAs downstream of lin-14 (which appears to regulate their transcription (Tsialikas et al (2017) Genetics 205: 251-262)), and upstream of both lin-28 and hbl-1.

Reviewer #2: The investigation of interactions between dauer regulatory and heterochronic pathways in C. elegans has revealed important insights into general mechanisms through which animals maintain developmental plasticity in response to environmental stress. In this manuscript the authors elucidate a new function for the FoxO transcription factor DAF-16 in the maintenance of epidermal stem cell fate and quiescence during dauer diapause. Building on prior work showing that DAF-16 is required for the re-establishment of multipotency in the vulval precursor cells during dauer arrest, the authors use a col-19p::GFP reporter, a marker of adult epidermal seam cell fate, to show that dauers with reduced daf-16 activity exhibit precocious col-19p::GFP expression. Using genetic and transcriptomic approaches, they demonstrate that daf-16 inhibits the expression of col-19p::GFP through the canonical heterochronic gene lin-41. While lin-41 is also required for col-19p::GFP repression in reproductively developing larvae, this work reveals that lin-41 acts through a distinct mechanism to inhibit col-19p::GFP expression in dauers, doing so in a manner that is independent of the downstream heterochronic gene lin-29.

This is a rigorously executed and convincing study that establishes DAF-16 as a dauer-specific heterochronic gene, uncovers a new role for DAF-16 in the execution stage of dauer morphogenesis, and reveals another example of the plasticity of developmental pathways in the context of environmental stress. This work will be of general interest to investigators interested in the molecular underpinnings of developmental plasticity.

There are several points that should be addressed in a revised manuscript:

1. Please include line numbers.

2. Introduction, second paragraph, second-to-last line: delete the Pierce et al. reference.

3. Throughout the manuscript, "DAF-16/FOXO" should be used to improve clarity for readers not familiar with C. elegans.

4. Introduction, second paragraph, last line: the two references listed address the regulation of DAF-16 subcellular localization, not target genes. I would consider citing a dauer review (e.g., Fielenbach and Antebi).

5. Results, first paragraph line 12 and second paragraph lines 3-4: Stating that daf-16 and daf-5 mutant larvae are "dauer-defective" could be confusing to non-mavens. I would consider being more specific to enhance clarity, e.g., "As larvae with reduced daf-16 activity bypass dauer arrest in the context of reduced DAF-2 insulin signaling,..." and an analogous sentence for daf-5.

6. Figures 1D and S2: Since the col-19p::GFP expression phenotypes of daf-16(0) and daf-16a/f mutants do not differ in terms of penetrance (1D) but do appear to differ in terms of GFP expression levels per cell and number of cells/animal, both expression levels and number of GFP+ cells in these two mutant backgrounds should be quantified.

7. Paragraph entitled "daf-16 promotes expression of lin-41 during dauer," lines 10-11: Similar to 6., GFP expression in lin-41(RNAi) and daf-16(RNAi) should be quantified.

8. Figure 1E: what do "1" and "2" denote?

9. Figure 2, legend: consider changing "sisters" to "family members."

10. Figure 2B: it would be nice to include a daf-16 RNAi sample in this experiment so that the relative effects of daf-16 RNAi and lin-41 RNAi on col-19p::GFP expression could be visualized (differences in RNAi efficiency notwithstanding).

11. Figure S1, legend, line 4: "da Graca et al."

12. Figure S7: typo: "paralyzed"

13. Figure S8B: I don't think the 250ms exposure adds anything.

14. Table 1: allele names for the daf-16a mutants are "tm5030" and "tm5032."

Reviewer #3: Comments on PGENETICS-D-21-00989

Wirick and colleagues describe that daf-7(e1372); daf-16(null) dauers misexpress adult-specific collagen col-19 and RNA-binding protein lin-41 is required for this. Using RNA-sequencing comparing daf-7(e1372); daf-16(null) vs daf-7(e1372) dauers, they found an upregulation of about 20 collagens normally expressed in adulthood and a downregulation of about 20 collagens normally expressed during dauers. From this they conclude a loss of multipotency of the epidermal stem cells (seam cells) in these daf-7(e1372); daf-16(null) dauers potentially explaining this misexpression of adult-specific collagens.

Overall, the study is well designed and experiments were done with the highest standards. In general, I like the study and have only a few questions and minor issues that could be addressed.

I have several questions of things that were unclear to me and should be clairified:

1. Are these adult-specific collagens expressed in seam cells or hypodermis? If not in seam cells, how does this relate to loss of multipotency?

2. Is col-19 also misexpressed in daf-16(null) dauers induced by starvation? Or is this specific to the tgf beta pathway?

3.

Minor issues:

1. Manuscript could be more concise and less repetitive across sections (intro, results, discussion)

2. Compared to PMID: 9649524, how do the findings add conceptional novel insights? Also other genes like lin-42, lin-28, mir-41 etc that are upstream of lin-29 and regulate Pcol-19::GFP were not discussed or investigated.

3. The transcription factor lin-29 is not required. The authors mention 112 transcription factors from their RNA seq and suggest that they in the absense of daf-16 regulate misexpression of adult collagens or loss of mulitpotency. This could be easily tested by a small RNAi screen using Pcol-19::GFP as a read out. I am not saying they need to do this, but it would be nice to get some insight of a potential mechanism here.

**Have all data underlying the figures and results presented in the manuscript been provided?**

Reviewer #1: Yes

Reviewer #2: Yes

Reviewer #3: Yes

PLOS authors have the option to publish the peer review history of their article (what does this mean?). If published, this will include your full peer review and any attached files.

Reviewer #1: No

Reviewer #2: **Yes: **Patrick Hu

Reviewer #3: No

---

## [Decision Letter · Decision Letter 1]

15 Oct 2021

Dear Dr Karp,

We are pleased to inform you that your manuscript entitled "daf-16/FOXO blocks adult cell fate in Caenorhabditis elegans dauer larvae via lin-41/TRIM71" has been editorially accepted for publication in PLOS Genetics. Congratulations!

Yours sincerely,

Coleen T. Murphy

Associate Editor

PLOS Genetics

Gregory P. Copenhaver

Editor-in-Chief

PLOS Genetics

Comments from the reviewers (if applicable):

The reviewers are in agreement that the authors have addressed all of their critiques.

Reviewer's Responses to Questions

**Comments to the Authors:**

Reviewer #1: The revised manuscript admirably addresses all the reviewer critiques; publication of the paper in its current form in PLOS Genetics is recommended.

Reviewer #2: The authors have adequately addressed my critique of the original submission.

Reviewer #3: The authors have addressed all my previous comments.

**Have all data underlying the figures and results presented in the manuscript been provided?**

Reviewer #1: Yes

Reviewer #2: Yes

Reviewer #3: Yes

PLOS authors have the option to publish the peer review history of their article (what does this mean?). If published, this will include your full peer review and any attached files.

Reviewer #1: No

Reviewer #2: **Yes: **Patrick Hu

Reviewer #3: No

**Data Deposition**

http://datadryad.org/submit?journalID=pgenetics&manu=PGENETICS-D-21-00989R1

**Press Queries**

---

## [Editor Report · Acceptance letter]

4 Nov 2021

PGENETICS-D-21-00989R1 

*daf-16*/FOXO blocks adult cell fate in *Caenorhabditis elegans* dauer larvae via *lin-41*/TRIM71 

Dear Dr Karp, 

We are pleased to inform you that your manuscript entitled "*daf-16*/FOXO blocks adult cell fate in *Caenorhabditis elegans* dauer larvae via *lin-41*/TRIM71" has been formally accepted for publication in PLOS Genetics! Your manuscript is now with our production department and you will be notified of the publication date in due course.

With kind regards,

Zita Barta

PLOS Genetics

On behalf of:
